# Anaerobiosis favors biosynthesis of single and multi-element nanostructures

**Mirtha Ríos-Silva[1,2], Myriam Pérez[1], Roberto Luraschi[1], Esteban Vargas[3], Claudia Silva-Andrade[4], Jorge Valdés[4], Juan Marcelo Sandoval[5], Claudio Vásquez[1†], Felipe Arenas[1]***

1 Laboratorio de Microbiología Molecular, Departamento de Biología, Facultad de Química y Biología, Universidad de Santiago de Chile, Santiago, Chile, 2 Research Center on the Intersection in Plasma Physics, Matter and Complexity, $P^2$mc, Comisión Chilena de Energía Nuclear, Santiago, Chile, 3 Center for the Development of Nanoscience and Nanotechnology (CEDENNA), Santiago, Chile, 4 Centro de Genómica y Bioinformática, Universidad Mayor, Santiago, Chile, 5 Facultad de Ciencias, Universidad Arturo Prat, Iquique, Chile

† Deceased.
* felipe.arenass@usach.cl

**Data Availability Statement:** All relevant data are within the paper and its Supporting Information files. You can also go directly through the following link, in which all contigs are available with their

## Abstract

Herein we report the use of an environmental multimetal(loid)-resistant strain, MF05, to biosynthesize single- or multi-element nanostructures under anaerobic conditions. Inorganic nanostructure synthesis typically requires methodologies and conditions that are harsh and environmentally hazardous. Thus, green/eco-friendly procedures are desirable, where the use of microorganisms and their extracts as bionanofactories is a reliable strategy. First, MF05 was entirely sequenced and identified as an *Escherichia coli*-related strain with some genetic differences from the traditional BW25113. Secondly, we compared the CdS nanostructure biosynthesis by whole-cell in a design defined minimal culture medium containing sulfite as the only sulfur source to obtain sulfide reduction from a low-cost chalcogen reactant. Under anaerobic conditions, this process was greatly favored, and irregular CdS (ex. 370 nm; em. 520–530 nm) was obtained. When other chalcogenites were tested (selenite and tellurite), only spherical $Se^0$ and elongated $Te^0$ nanostructures were observed by TEM and analyzed by SEM-EDX. In addition, enzymatic-mediated chalcogenite (sulfite, selenite, and tellurite) reduction was assessed by using MF05 crude extracts in anaerobiosis; similar results for nanostructures were obtained; however $Se^0$ and $Te^0$ formation were more regular in shape and cleaner (with less background). Finally, the *in vitro* nanostructure biosynthesis was assessed with salts of Ag, Au, Cd, and Li alone or in combination with chalcogenites. Several single or binary nanostructures were detected. Our results showed that MF05 is a versatile anaerobic bionanofactory for different types of inorganic NS. synthesis.

## Introduction

Inorganic nanostructures (NS) have gained prominence in industries due to their adjustable physicochemical characteristics [1]. These materials can comprise metals or non-metals or

respective sequences: https://www.ncbi.nlm.nih.gov/nuccore/1679187305.

**Funding:** This work received financial support from FONDECYT (Fondo Nacional de Ciencia y Tecnología) Regular 1160051 (CV), National doctoral scholarship CONICYT (Comisión Nacional de Investigación Científica y Tecnológica) 21170508 (MR), support from USA1799 Vridei (Vicerrectoría de Investigación, Desarrollo e Innovación) 021943CV_GO Universidad de Santiago de Chile (MR, CV), Basal FB0807 CEDENNA (EV) and Centro de Genómica y Bioinformática, Universidad Mayor (JV) is also acknowledged.

**Competing interests:** The authors have declared that no competing interests exist.

take the form of an oxide, hydroxide, chalcogenide, or phosphate compound [2]. Currently, biosynthetic nanostructures biosynthesized by cells and bacteriophages cover at least 55 elements in the periodic table, which involves 146 single-element and multi-element NS [3].

When exposed to metal and non-metal ions, microorganisms transport these ions into cells through membrane carriers. Alkali and alkaline earth metal cations are used by cells to maintain intracellular homeostasis, serve as cofactors in enzymatic reactions, and survive. However, the presence (above trace level) of transition-metal ions such as $Cu^{2+}$ and $Ag^{+}$, as well as post-transition metal ions like $Zn^{2+}$, $Cd^{2+}$, and $Hg^{2+}$, are toxic. Thus, cells have developed strategies to remove these toxic ions either by ion export using efflux pumps or by other protective mechanisms, such as the reduction of inorganic ions to their elemental forms [4].

Surprisingly, certain microorganisms can grow in the presence of different metal(loid)s, leading to multi-metal(loid) resistance (MMR). This phenotype is interesting when microorganisms are considered as possible factories to produce inorganic nanomaterials. In this line, bacteria-producing metal nanomaterials with antimicrobial properties are a realistic biotechnological promise toward sustainability [5]. Bacteria reduce metal ions, chalcogens oxyanions, or other elements using NADH- or NADPH-dependent reductases, catalases, and terminal oxidases or by non-enzymatic routes such as thiol groups, and phosphates, among others [6–9].

Chalcogens are of special interest since sulfur, selenium, and tellurium can generate semiconductor compounds [10]. A common aspect of chalcogens is their ability to form oxyanions in oxidation states +4 and +6, known as chalcogenites and chalcogenates. Of the three chalcogenites, the most complex to approach -from the biological point of view- is tellurite. In contrast, sulfite and selenite are essential and have well-known biological functions [11]. Moreover, chalcogens can often be in their elemental/zero-state (0) or as chalcogenides (-2) associated with metals, which are of great technological interest, e.g., chalcogenide-based Quantum Dots (QDs). Most studies on chalcogenite bioreduction and NS formation, have been carried out via four electrons of selenite or tellurite to their elemental forms $Se^{0}$ and $Te^{0}$, respectively [12–16]. Moreover, some bacteria contain enzymes that carry out physiological reductions of 6 electrons, such as nitrate reductase or sulfite reductase [17].

Regarding nanobiosynthesis, CdS QDs have been synthesized in *E. coli* by directly administering $H_2S$ and $CdCl_2$ [18], or other species such as *Stenotrophomonas maltophilia* dealing with cysteine and $Cd(CH_3COO)_2$ [19] or *Acidithiobacillus thiooxidans* ATCC 19703 using $S_8$, $CdCl_2$, and glutathione at pH 2.3 [20]. On the other hand, CdSe or CdTe QDs biosynthesis has also been reported [21–24]. However, precursors usually used i.e. $H_2S$ or cysteine, aren't cost-effective for larger-scale processes. Moreover, the electrical, optical, and catalytic properties of inorganic NS are dependent on their elemental composition, crystallinity, size, and shape. It is, thus, important to be able to experiment with which types of inorganic NS can be biosynthesized in a specific biological model, so we can rationally prepare NS for a given application [3]. Besides, considering that toxicity mechanisms are normally associated with oxidative stress, an interesting approach for NS biosynthesis is anaerobiosis.

In this study, we evaluated the MF05 strain as a bionanofactory under anaerobiosis. MF05 has an MMR phenotype and was able to synthesize single- and multi-element nanostructures such as CdS, $Se^{0}$ and $Te^{0}$ Ag, Au, Cu, Li, Se, Te, and NS including CdS, Li-S, Ag-S, Au-Se, Cu-Te, Ag-Te, and Cd-Se. This was based on its ability to reduce chalcogenites without electron leakage (under the absence of molecular oxygen) and its resistance to toxic metal ions. In this line, the MF05 strain was previously studied in the presence of oxygen, with MICs for gold, copper, selenite, and tellurite, which were 0.25, 6.25, 500, and 1 mM, respectively. These values are significantly higher than those of *E. coli* BW25113, which are 0.16, 1, 125, and 0.004 mM for gold, copper, selenite, and tellurite, respectively. Therefore, MF05 had a resistant

phenotype and displayed the ability to synthesize metal NS of Ag and Au in the presence of molecular oxygen, but it was not able to reduce tellurite or selenite to their elemental forms [25]. We hypothesized that by using whole cells or parts of them in the form of crude extracts under anaerobic conditions, we would be able to avoid leakage of electrons that occurs naturally in the process of reducing $O_2$ as an electron acceptor, and thus generate a more reducing environment.

## Materials and methods

### Bacterial strains and culture conditions

MF05 was previously aerobically characterized by Figueroa et al. 2018 [25] and *E. coli* BW25113 was used as a wild-type control bacterium. Cells were grown routinely at 37°C in either LB, M9 minimum, or modified M9 minimum (M9*) medium containing 1 mM $Na_2SO_3$ and 1 mM $MgCl_2$ (instead of $MgSO_4$). Growth in liquid medium was generally started with a 1% dilution of overnight grown cultures. For experiments under anaerobic conditions, a Coy anaerobic chamber (Coy Laboratory Products, Inc. Grass Lake, MI, USA) with a 100% $N_2$ internal atmosphere was used.

### Growth curves

Every 30 minutes, the anaerobic and aerobic growth curves were spectrophotometrically monitored in two TECAN M100 Pro multi-plate readers, one of which was located inside the anaerobic Coy chamber.

### Biogenic $H_2S$ determination

$H_2S$ release was evaluated as described by Narayanaswamy and Sevilla [26]. Cultures of 3 ml inoculated at 1% were grown in M9 or M9* medium in the presence or absence of $O_2$. $H_2S$ was measured on filter papers that had been immersed in 0.1 M $Pb(CH_3COO)_2$ for 1 h and then dried at 60°C. These were placed on the top of the tube of each culture and were sealed using Breatheeasy® membranes to allow the gas passage without affecting the kind of respiration that was being evaluated. The formation of black precipitates in the filter paper corresponded to lead sulfide (PbS). Filters were scanned at high resolution and analyzed with the ImageJ software.

### Preparation of cell extracts

Starting with a 1% inoculum, cultures of 500 ml were grown for 12 h in anaerobiosis. They were collected in 50 ml Falcon tubes, covered, and centrifuged at 9,000 x g for 10 min at 4°C. Cells were suspended in a 50 mM Tris-HCl buffer (pH 7.4), supplemented with 0.1 mM PMSF. Cells were sonicated on ice inside the Coy chamber with four pulses of 20 s. Centrifugation at 9,000 x g for 15 minutes at 4°C was used to remove cell debris. The supernatant, containing soluble proteins, was considered the crude extract.

### Enzymatic activity measurements

The reducing activities for chalcogenites present in crude extracts were analyzed in anaerobiosis at pH 9.0 because there was less reduction of controls without extracts. The reactions were carried out in a final volume of 200 μL and monitored spectrophotometrically with a TECAN M100 Pro multi-plate reader based on the specific conditions for each chalcogenite:

For sulfite: 5 mM $Na_2SO_3$, 0.5 mM NAD(P)H, 0.5 mM β-mercaptoethanol (2-ME), and 25 μL of crude extract and buffer were used in the reaction. NAD(P)H consumption at 340 nm was measured every 30 s for 5 min.

For selenite: 1 mM $Na_2SeO_3$, 0.5 mM NAD(P)H, 0.5 mM 2-ME, and 25 μL of crude extract and buffer were used in the reaction. Measurements were made at 400 nm, whose peak represents elemental selenium formation ($Se^0$), every 30 s for 5 min.

For tellurite: 1 mM $Na_2TeO_3$, 0.5 mM NAD(P)H, 0.5 mM 2-ME, and 25 μL of crude extract and buffer were used in the reaction. Measurements were made at 500 nm, whose peak represents elemental tellurium formation ($Te^0$), every 30 s for 5 min.

An enzyme unit (U) was defined as the amount of enzyme needed to increase (and decrease in the case of sulfite) by 0.001 units of absorbance in 1 min.

## NS synthesis and purification

**Cell cultures NS synthesis.** For NS synthesis and characterization, sulfite concentrations determined by the whole-cell system in a checkerboard format were considered for CdS formation, which corresponded to 864 μM $CdCl_2$ in the presence or absence of $Na_2SO_3$ up to 14.58 mM (sulfite incorporated into the medium). For both, selenite and tellurite, the concentration used was 1 mM. The concentrations used for lithium, gold, silver, and copper salts were 125 mM, 500 μM, 4 mM, and 4 mM, respectively.

**Purification of whole-cell synthesized NS.** Cultures of 200 mL were grown to a stationary phase to be treated with cadmium and chalcogenites. Cells were collected in 50 mL Falcon tubes at 9,000 x g for 10 minutes at 4°C, then suspended in 1 ml of 50 mM Tris-HCl buffer pH 7.4. Samples were sonicated on ice with 4 pulses of 1 min. In parallel, one day before the ultracentrifugation, a sucrose gradient was made with cushions of 20, 40, 50, and 60% in 5 ml tubes so that they were previously equilibrated overnight at 4°C. Samples of 800 μl were deposited on top of gradients and centrifuged at 300,000 x g for 2 h in a swinging-bucket rotor without brake. After centrifugation, tubes were observed with a UV transilluminator to detect those fractions displaying fluorescence. Around 25 fractions (200 μL each) were collected for future characterizations.

*In vitro* **NS synthesis.** From the enzymatic activity results, MF05 crude extracts were obtained to make checkerboards with NADH or NADPH for $CdCl_2$ with $Na_2SO_3$, $Na_2SeO_3$, or $Na_2TeO_3$, which were tested for 24 h at 37°C with constant agitation in anaerobiosis. For the assay, in a final volume of reaction of 200 μL, 0.5 mM NAD(P)H, 0.5 mM 2-ME and crude extracts (25 μl) were combined with $CdCl_2$: 0, 54, 108, 215, 432, or 864 μM; $Na_2SO_3$: 0, 0.5, 1 or 5 mM; and $Na_2SeO_3$ or $Na_2TeO_3$: 0, 0.5,1 or 2 mM, and glycine buffer pH 9.0. After treatments, samples were visualized under visible or UV light. Then, excitation and emission fluorescence scan spectra were carried out to choose optimal concentrations of cadmium and/or chalcogenites and select them for electron microscopy characterization.

## Fluorescence of biosynthesized QDs

Samples were observed in a UV transilluminator (λex 254 nm) to determine the wavelength at which they emitted. Subsequently, an approximated emission wavelength was set to make an excitation scan. Finally, the wavelength of higher excitation was selected to scan the emission and determine precisely the emission wavelength. These determinations were carried out using the TECAN Infinite M200 Pro fluorimeter.

## Transmission electron microscopy (TEM)

The NS generated from cell cultures were observed using a Hitachi HT7700 transmission electron microscope. Each sample was prepared by depositing ~20 μl of NS suspension in a copper

grid (200 mesh) using a continuous carbon film or lacey carbon. The analyses were carried out at the "Center for the Development of Nanoscience and Nanotechnology"—CEDENNA, USACH.

## Determination of the chemical composition of NS by X-ray energy dispersion spectroscopy (EDX)

Approximately 20 μL of samples containing NS, as well as their respective controls, were deposited on a conductive carbon adhesive on a support pin. Scanning electron microscopy (SEM) analyses were performed using a Zeiss EVO MA-10 microscope with a tungsten filament gun and an energy dispersive X-ray (EDX) spectrum. Data was collected using an Oxford instrument X-ray system (connected to a microscope equipped with a Penta FET precision detector). Images were taken from the samples at an acceleration voltage of 20 kV and 8 mm working distance. Analyses were carried out at the "Center for the Development of Nanoscience and Nanotechnology", CEDENNA, USACH.

## Sequencing, annotation, and comparative genomics

Genomic DNA from MF05 was extracted using Wizard® Genomic DNA Purification Kit (Promega). The QuantiFluor fluorometer was used to calculate concentration. Libraries were prepared with TruSeq Nano DNA Library Prep Kit (Illumina). For quality control of DNA libraries, size, distribution, and integrity of fragments were analyzed with 2100 Bioanalyzer Instrument (Agilent) with an average size of 407 bp. Sequencing was carried out using Illumina HiSeq. Contig assembly was compared by BlastX with the NR, Uniref100, and SwissProt databases. From the annotation, a comparison was made with *E. coli* K-12 BW25113 by Cluster of Orthologous Groups (COGs).

## Data analysis

Plots and statistical analyses were carried out using GraphPad Prism 6.0 (GraphPad Software, Inc.) or Excel (Microsoft Office 365, Microsoft Corporation). Statistical analysis was used considering significance as follows: *, $p < 0.05$, **, $p < 0.01$, ***, $p < 0.001$ and ns, no significant difference.

# Results

## Identification of MF05 strain

MF05 is an environmentally isolated strain with an MMR phenotype. So firstly, the whole genome of MF05 was sequenced. Respective assemblies were carried out and it was found that it displayed size of 4,591,446 bp and 266 contigs, the longest being 414,005 base pairs. From this data, 4,389 genes were predicted, of which 4,312 hits against the nr database were detected when performing BlastP, and of these, 1,930 were hits for Proteobacteria, 1,608 for *E. coli*, and 522 for Enterobacteriaceae. Additionally, BlastX was performed against the nr and UniRef100 databases, in which 9,629 and 7,366 hits were displayed, respectively. When comparing the genome of MF05 with databases, the hits were mainly with *E. coli* and, to a lesser extent, with *Shigella* and Proteobacteria. Then, an ANI (Average Nucleotide Identity) analysis was performed among several strains such as *Acinetobacter schindleri*, *Salmonella bongori*, *S. enterica*, *Shigella sonnei*, *S. dysenteriae*, *S. flexneri*, six different strains of *E. coli*, and MF05. By previous bioinformatic analyses, a heatmap showed that MF05 showed identity with other strains of *E. coli* K-12 (Fig 1). Through ANI, it could be confirmed that the closest resemblance of MF05 is

**Fig 1. Heatmap of ANI analysis for MF05 against 12 different proteobacteria strains.** Nucleotide identity comparison between *Acinetobacter schindleri* ACE, *Salmonella bongori* NCTC 12419, *S. enterica* serovar Typhi CT18, *Shigella sonnei* BS1058, *S. dysenteriae* Sd197, *S. flexneri* 301, *E. coli* BW25113, *E. coli* MG1655, *E. coli* W3110, *E. coli* BL21 (DE3), *E. coli* O157:H7 Sakai, *E. coli* O104:H21 CSFSAN00236 and MF05.

to *E. coli*. However, COG function analysis informed us that there are differences between MF05 and BW25113 (S1 Fig).

## Whole-cell synthesis of chalcogen-based NS under anaerobiosis

For nanostructure biosynthesis, the M9 medium avoids interferences during nucleation and growth processes. However, this medium contains magnesium sulfate, in which the sulfur oxidation state is +6. Therefore, M9 was modified, and magnesium sulfate was replaced by magnesium chloride and sodium sulfite, in which the sulfur oxidation state is +4 (hereafter M9*). Growth curves comparing M9 and M9* were assessed for MF05 and *E. coli* BW25113, both in aerobiosis and anaerobiosis, in which there was identical growth behavior in both media (S2 Fig). After that, checkerboards with increasing concentrations of sulfite and cadmium were carried out in M9 and M9* medium in aerobiosis and anaerobiosis. It was observed that in both media, *E. coli* BW25113 and MF05 did not show fluorescence when exposed to sulfite and cadmium in aerobiosis (Fig 2A, 2C, 2E and 2G). Conversely, anaerobic cultures in M9* medium showed significant fluorescence, both in *E. coli* BW25113 and MF05 (Fig 2F and 2H), indicating that this medium promotes the formation of CdS QDs, with CdCl$_2$ 864 µM and it does not require additional sulfite to that already contained in the medium.

To assess if M9* medium favors CdS QDs formation through a higher release of sulfide compounds, such as H$_2$S, a headspace detection test for MF05 was carried out with lead acetate using M9 or M9* media under aerobic and anaerobic conditions. In the absence of oxygen, greater PbS formation occurred when MF05 grew in M9*, which also happened, although to a lesser extent, in aerobiosis (Fig 3A). Thereafter, the time influence on fluorescence coming from CdS QDs formation was determined. To do so, the MF05 strain was grown in M9* to a stationary phase and treated with CdCl$_2$, as previously determined on checkerboards. Then, samples were collected several times post-treatment, and all samples displayed a similar fluorescence pattern, which decreased considerably at 24 h (Fig 3B). To determine at which wavelength CdCl$_2$-treated cultures were being excited and at which wavelength they emitted,

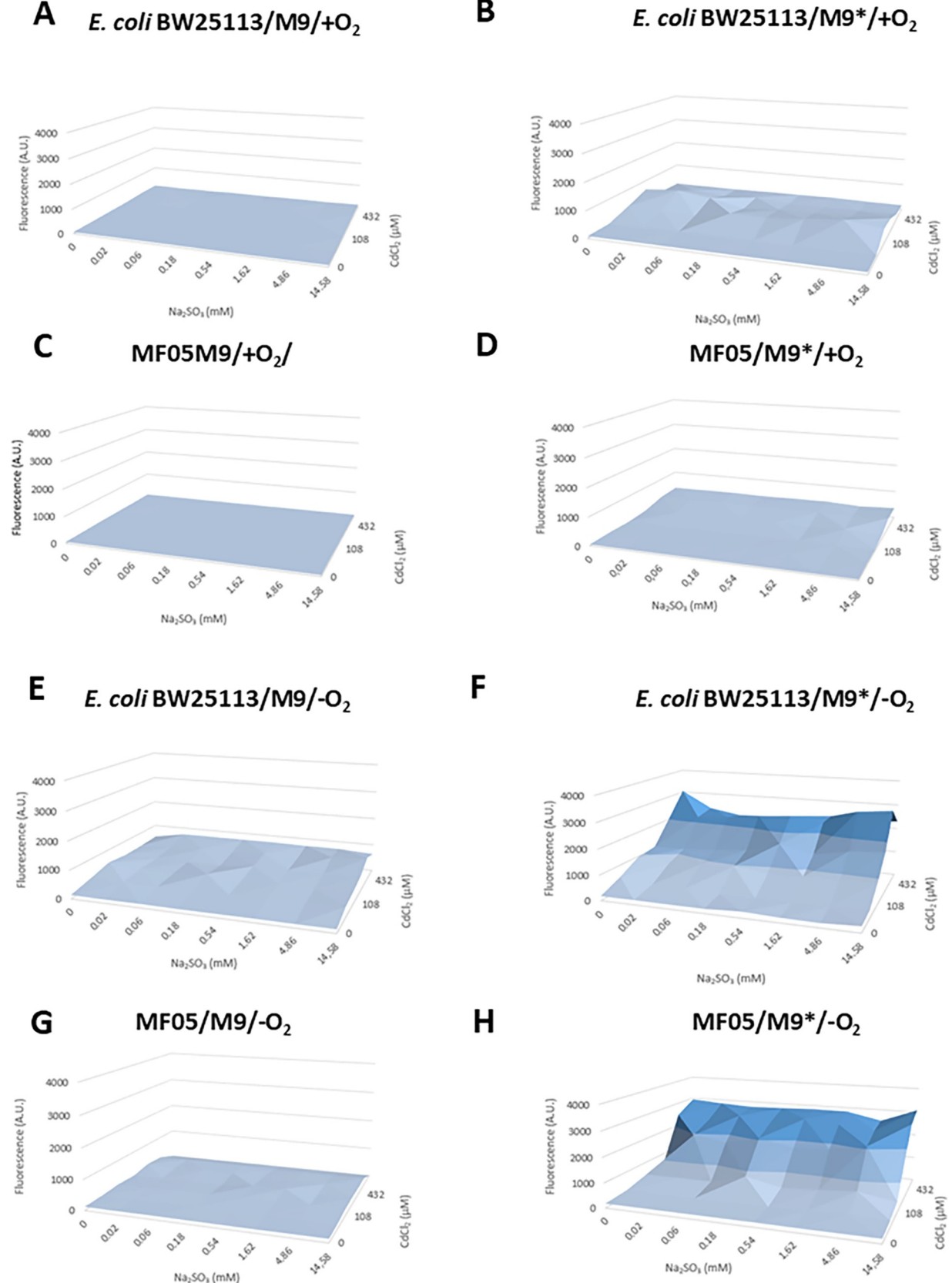

**Fig 2. Fluorescence emission of sulfite and cadmium checkerboards to check CdS QDs formation by MF05 cells with M9 or M9\* medium under aerobic or anaerobic conditions.** Aerobic sulfite + $CdCl_2$ checkerboards assays for *E. coli* BW25113 grown in (A) M9 or (C) M9\* medium, and MF05 grown in (E) M9 or (F) M9\* medium. Anaerobic sulfite + $CdCl_2$ checkerboards assays for *E. coli* BW25113 in (B) M9 or (D) M9\* medium, and MF05 grown in (F) M9 or (H) M9\* medium. Fluorescence was monitored with 230 and 640 nm of excitation and emission wavelength, respectively. The results represent the average of 3 independent trials.

excitation and emission scans of treated and untreated cells were performed. Cadmium-treated cells showed maximum excitation at 370 nm at all times tested. The highest peak observed was at 30 min, followed by 2 h (Fig 3C). Regarding emission, peak values ranged between 520 and 530 nm, except for 24 h samples, in which emission was lost (Fig 3D).

Once the time and compound concentrations were set, the CdS QDs biosynthesized samples were analyzed by electronic microscopes, TEM, and SEM. An abundant pattern of dense electron points was observed that represent irregular CdS QDs (Fig 4B), as compared with the untreated sample (Fig 4A). Regarding the chemical composition analysis as determined by EDX, in the untreated sample, the most prevalent element was carbon, followed by oxygen, sodium, chlorine, sulfur, phosphorus, and potassium. Meanwhile, treated samples showed all

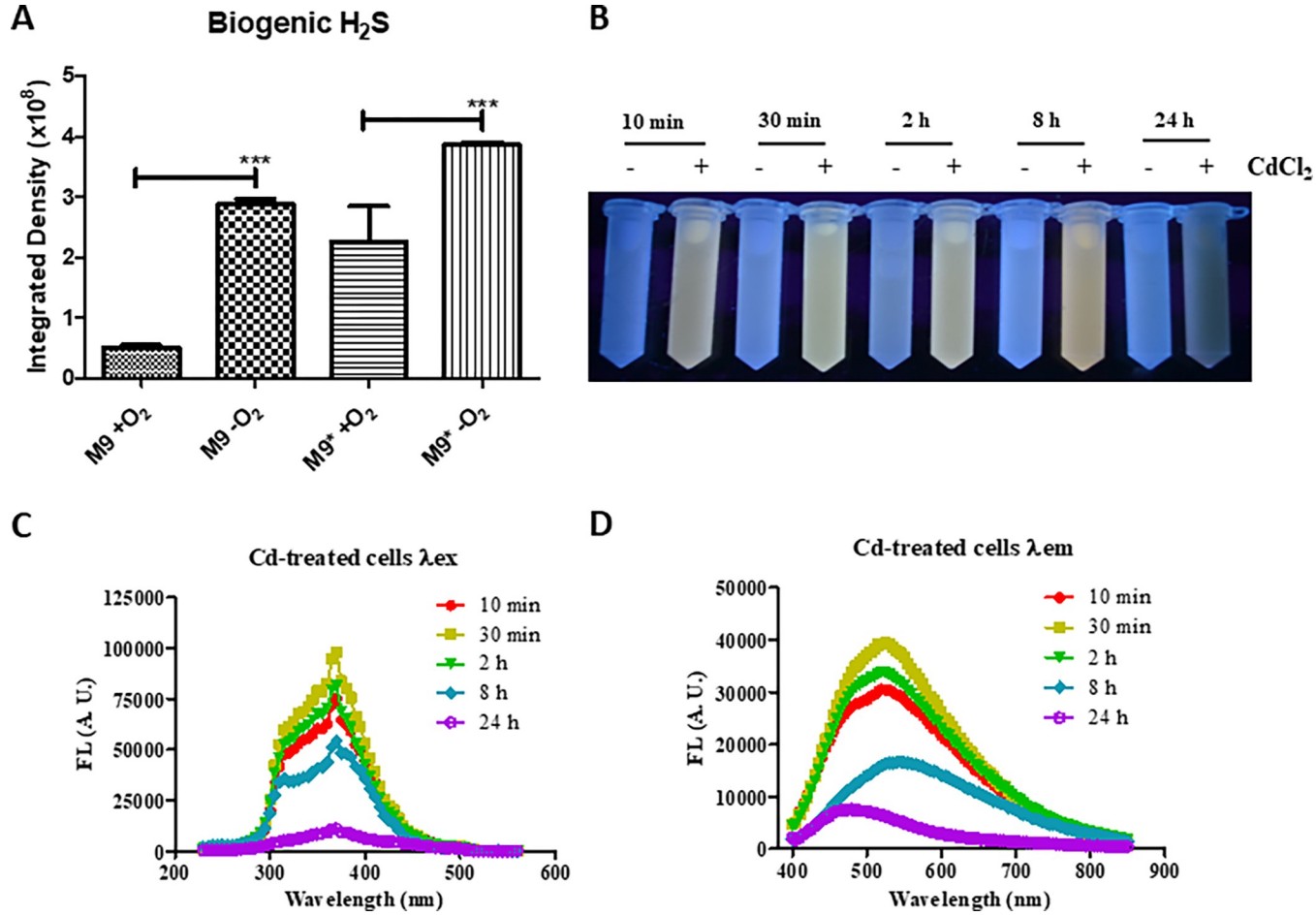

**Fig 3. Release of $H_2S$ by MF05 in M9 or M9\* media, under aerobic or anaerobic conditions, and fluorescence tracing of CdS QDs formation at different times under anaerobic conditions.** (A) Integrated density of image processing of $H_2S$ release by MF05 cells grown in M9 or M9\* medium under aerobic or anaerobic conditions with PbS filter papers. (B) CdS QDs formation after anaerobic growth of MF05 in M9\* medium treated with 864 μM $CdCl_2$ with their respective negative controls. (C) Excitation and (D) emission spectra scan of Cd-treated MF05 cells. λex: excitation with 590 nm emission wavelength; λem: emission with 370 nm excitation wavelength. For (A) the results represent the average of three independent trials ± SD \*\*\*, P < 0.001.

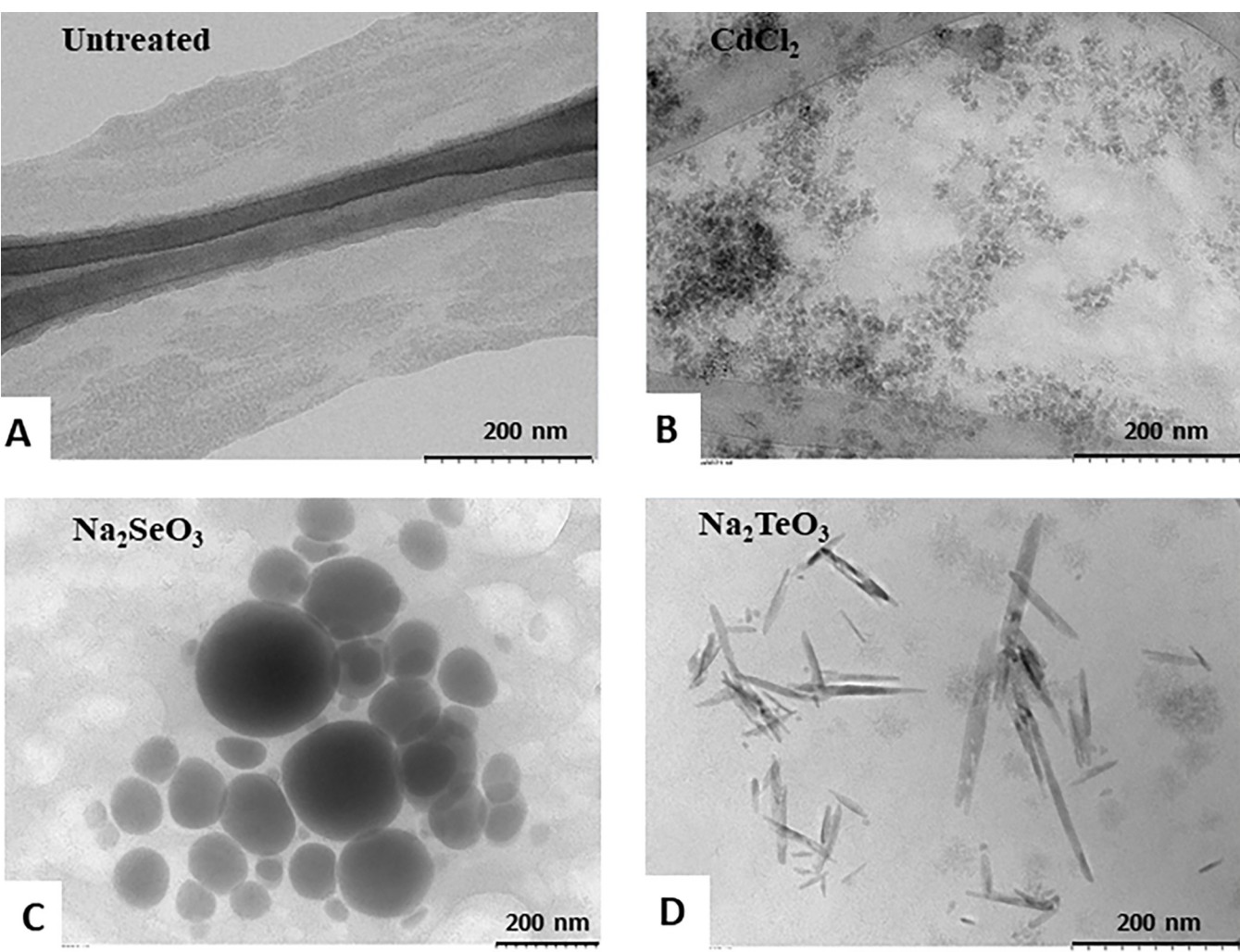

**Fig 4. TEM NS of CdS QDs, Se, or Te by MF05 cells.** Synthesis of NS by MF05 whole-cell in anaerobiosis (A) control and after treatments with (B) CdCl₂; (C) Na₂SeO₃ or (D) Na₂TeO₃.

the cellular elements and cadmium (S3 Fig). Given these results, we wanted to study the formation of other cadmium chalcogenides QDs (ex. CdSe and CdTe). However, in anaerobiosis, treatments with selenite or tellurite did not allow the clear formation of these QDs because the highest fluorescence exhibited at the lowest chalcogenite concentrations represented most probably CdS formation instead (S4 Fig). Despite being unable to synthesize whole-cell CdSe or CdTe QDs under these conditions, the formation of Se⁰ and Te⁰ NS by MF05 using M9* medium in anaerobiosis was carried out. It was observed that when exposed to selenite, electrodense elementary NS were formed with roughly spherical morphology (Fig 4C). In addition, tellurite treatment gave rise to an irregular, elongated NS (Fig 4D). In both cases, composition analysis showed the presence of chalcogens in addition to other organic elements (S3 Fig).

### *In vitro* synthesis of metal or chalcogen-based NS by MF05

As a cleaner alternative, *in vitro* synthesis was also considered. For these purposes, the reducing activities of chalcogenites of S, Se, and Te were examined. These assays were performed by

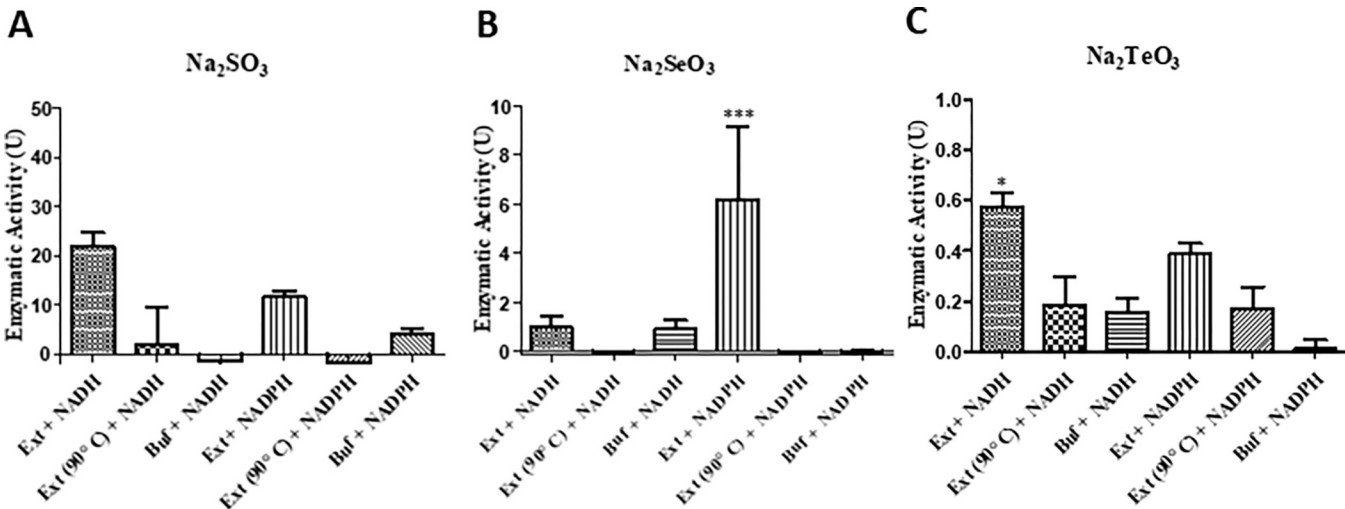

**Fig 5. Chalcogenite- reducing activity assays in crude extracts of MF05 in anaerobiosis.** Enzymatic activity by MF05 crude extracts for (A) sulfite reduction through NADH or NADPH consumption, (B) selenite reduction, through $Se^0$ formation, or (C) tellurite reduction through $Te^0$ formation. The results represent the average of three independent trials ± SD. *, $P < 0.05$; *** $P < 0.001$.

using crude extracts from MF05 cultures grown until the stationary phase in anaerobiosis. Controls contained only buffer or heated (90˚ C) crude extracts (as denatured proteins). The highest reducing activities for sulfite and tellurite were observed in the presence of NADH (Fig 5A and 5C), while for selenite, the best electron donor was NADPH (Fig 5B).

Then, cadmium-sulfite checkerboard assays were made using crude extracts with NADH or NADPH as electron donors. It was observed that when extracts were treated with NADH, 5 mM $Na_2SO_3$, and $CdCl_2$ at 108, 216, 432, or 864 μM, there was fluorescence corresponding most probably to CdS QDs (S5 Fig). Complementing the above, samples were observed by TEM and analyzed by composition with SEM-EDX. No NS was observed in any of the controls (S6 Fig); NS formation upon co-treatment with cadmium and sulfite was evidenced (Fig 6A). To determine whether it was possible to synthesize CdSe or CdTe QDs *in vitro* using MF05 extracts, new checkerboards assays of $Na_2SeO_3$ + $CdCl_2$ or $Na_2TeO_3$ + $CdCl_2$ were constructed with NADH or NADPH. However, consistent with the previous results in whole-cell assays, treatment with selenite or tellurite allowed the formation of spherical NS corresponding to $Se^0$ or elongated NS corresponding to $Te^0$ (Fig 6B and 6C). In addition, anaerobic assays with additional metals were performed in which $Ag^0$, $Au^0$, and $Cu^0$ NS were produced (Fig 6D and 6F). It should be noted that results of biosynthesis from whole-cell, as determined by SEM-EDX for $Se^0$ and $Te^0$, showed that their presence in the samples was 0.8% and 0.3%, respectively. In contrast, EDX results for $Se^0$ and $Te^0$, for *in vitro* biosynthesis, exhibited values of 9.8% and 22.1%, respectively (S7 Fig). This comparison makes it possible to affirm that *in vitro* systems allow higher efficiency for synthesizing NS of zero-state chalcogens, in addition to lower complexity for purification.

Finally, an *in vitro* combination of metals and chalcogenites treatments was performed using crude extracts of MF05 in the anaerobiosis. Firstly, despite the high reactivity of lithium or lithium with sulfite, NS was formed (Fig 7A and 7B). Interestingly, the combination of silver + sulfite, gold + selenium, copper + tellurite, and silver + tellurite showed NS with similar patterns in which there are different areas of electron-density inside the same NS (Fig 7C–7F). These results suggest that the anaerobic conditions favor multi-element NS formation. Even though we could not detect fluorescence in samples exposed to cadmium + selenite or

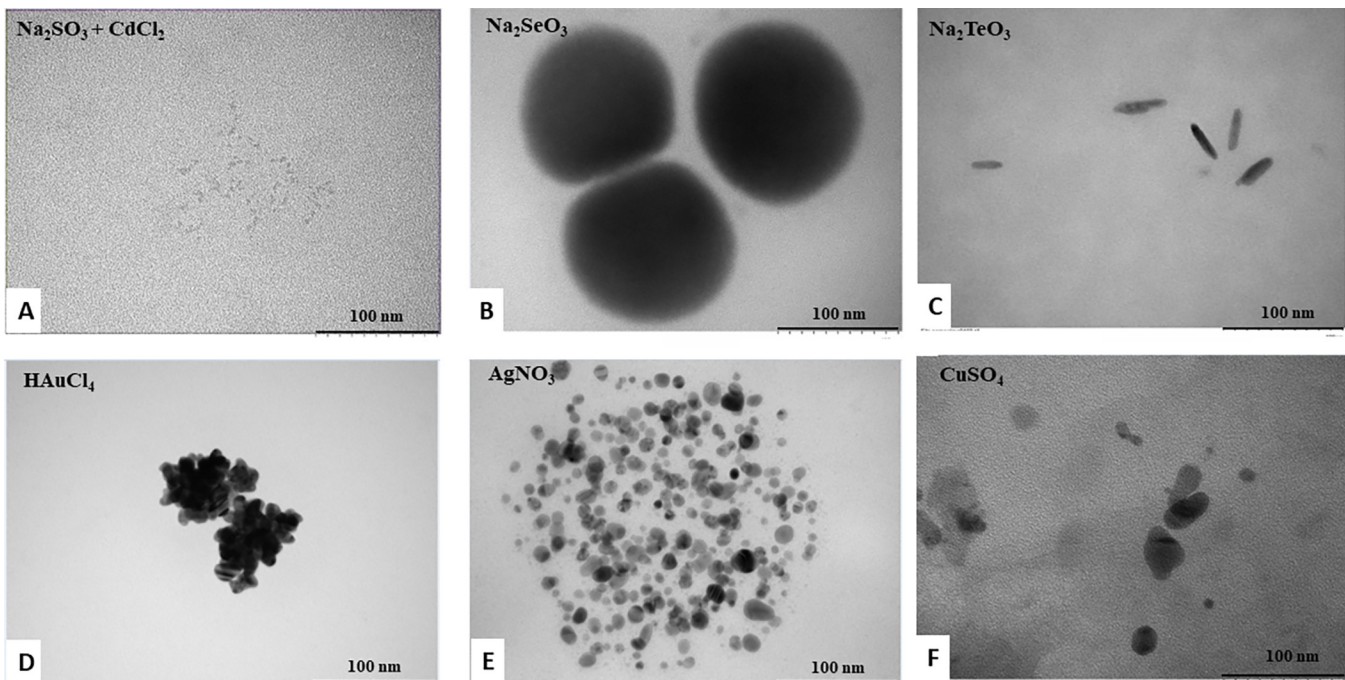

**Fig 6. TEM of NS from *in vitro* synthesis by crude extracts of MF05.** NS obtained after treatment with (A) $Na_2SO_3$ + $CdCl_2$; (B) $Na_2SeO_3$; (C) $Na_2TeO_3$; (D) $HAuCl_4$; (E) $AgNO_3$ or (F) $CuSO_4$.

cadmium + tellurite by whole-cells or *in vitro* synthesis. Thus, it is plausible that the alternative of bigger size NS in which the fluorescence properties of QDs can no longer be present. On one hand, treatment with selenite + cadmium showed very few single NS, which could be $Se^0$ or CdSe (Fig 7G). On the other hand, treatment with tellurite + cadmium evidenced two different shapes of NS; one was like that obtained before by whole-cell or *in vitro* synthesis with tellurite treatment, and the other with a spherical shape (Fig 7H). This result suggests that the absence of fluorescence in this sample was mainly due to $Te^0$ formation.

## Discussion

As MMR microorganisms are an interesting target for NS synthesis as a response mechanism to the toxicity of different elements, this work studied reduction as a tool for the elaboration of inorganic (mainly chalcogen)-based nanomaterials such as QDs or elementary NS by the MF05 strain. The main disadvantage of using aerobic microorganisms is the toxicity of metal (loid)s, which is frequently associated with ROS formation [4]. One of the major challenges for chalcogenide-based NS generation is the reduction from chalcogenites (+4: $SO_3^{2-}$, $SeO_3^{2-}$, $TeO_3^{2-}$) to chalcogenides (-2: $S^{2-}$, $Se^{2-}$, $Te^{2-}$). Of the three chalcogens, tellurium is the most difficult to work with because it displays the lowest electronegativity and the highest toxicity to cells. As for tellurium and selenium, several studies have mainly focused on elemental (zero-state) NS formation under aerobic conditions [12, 14, 25]. Biologically, chalcogenite oxidation state has been observed in volatile compounds such as hydrogen sulfide, dimethyl sulfide, dimethylselenide, and dimethyltelluride [11, 27, 28]. Thus, QDs formation was considered as a six-electron reduction screening approach; from chalcogenite (+4) to chalcogenide (-2).

Since culture medium is essential for all bioprocesses, it should be noted that LB medium contains metals such as Mg, Ca, Fe, and Zn at micromolar levels and Co, Ni, Mn, Mo, and Cu

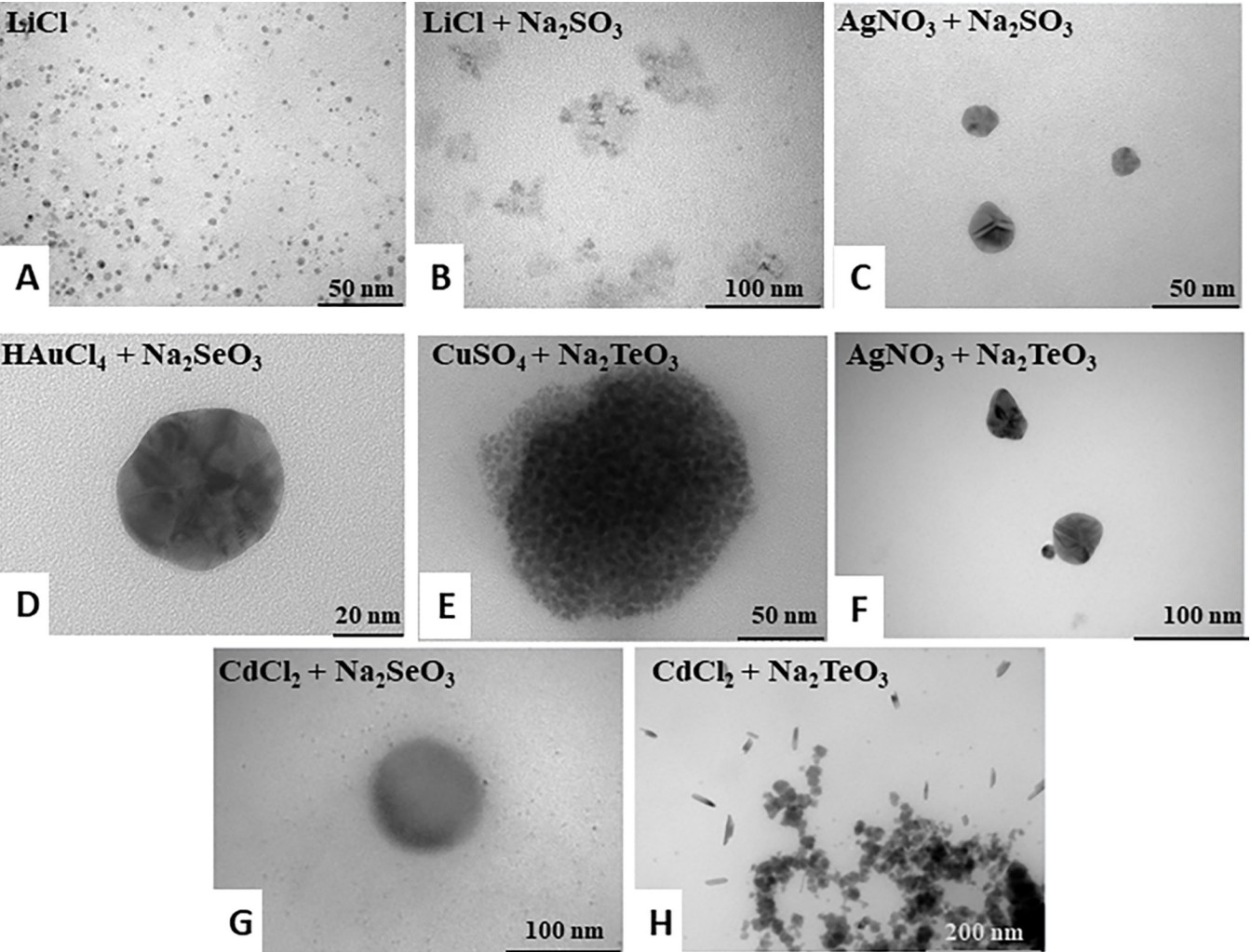

**Fig 7. TEM of NS from *in vitro* synthesis by crude extracts of MF05.** NS obtained after treatment with (A) LiCl; (B) LiCl + Na$_2$SO$_3$; (C) AgNO$_3$ + Na$_2$SO$_3$; (D) HAuCl$_4$ + Na$_2$SeO$_3$; (E) CdCl$_2$ + Na$_2$SeO$_3$; (F) CdCl$_2$ + Na$_2$TeO$_3$; (G) CuSO$_4$ + Na$_2$TeO$_3$; (H) AgNO$_3$ + Na$_2$TeO$_3$.

at nanomolar levels, and it contains an estimated amount of cysteine of ~400 μM, which can be associated with copper, making it less available for cell entrance. This phenomenon could occur not only with copper but also with other divalent elements [29]. Due to the above, it was considered necessary to study bioreduction in a defined M9 minimal medium. Previous work showed that the high phosphate content of M9 medium favors the formation of cadmium phosphate by decreasing its solubility and rendering it less accessible to the cell. This more controlled entry of cadmium would favor subsequently formed CdS QDs [20]. Furthermore, previous investigations have used operationally expensive precursors or additional elements added to the medium, which decreases the cost-effective feasibility of NS synthesis [18–20]. Given that *E. coli* can synthesize CdS QDs under certain conditions [18], sulfite was used as a precursor because it is economical and is part of several metabolic pathways.

From group 16 of the Periodic Table, after oxygen, sulfur is the main chalcogen used by cells. For instance, sulfate assimilation mechanisms. However, there is another series of sulfur-based compounds known generically as reactive sulfur species (RSS), defined as "redox-active molecules that contain sulfur and that can, under physiological conditions, oxidize or reduce

biomolecules," including thiols, disulfides, sulfenic acid, thiosulfinate, thiosulfonate, thiyl radicals, and $H_2S$ [30]. In this line, the biological system to obtain sulfides from sulfites or sulfates is further complicated. However, we can assert that more sulfides are being obtained given the presence of CdS QDs. Then, an interesting proposal would be to determine the efficiency within biological systems to analytically quantify the sulfite that is being reduced to sulfur, as well as what other sulfur-based precursors that are cost-effective could favor the observed phenotype.

In terms of bionanofactories, the cell itself generates confinement and energy change, which allows an open system with phase changes of solutions to precipitate that can allow the removal of products, such as NS, and thus favor their production. Regarding metal(loid)-chalcogenide synthesis, non-stoichiometric relationships favor the formation of more active axes within metal chalcogenide structures [31]. Chemical platforms, on the other hand, could not be extrapolated to a biological environment due to their high complexity and the multiple simultaneous processes that occur throughout the cell system. Besides, since sulfur is essential for life, it is not possible to remove it in cultures to test the other chalcogens. Therefore, it is likely that the product of chalcogen-based synthesis causes a mix between sulfur and selenium or tellurium. However, something remarkable in this whole-cell system was the stability of the particle size, since QDs kept their fluorescence properties for at least 10 weeks at 4° C. Fluorescence is directly related to particle size, so the bioprocess itself may confer biostabilizers to the particles, which could be very interesting to characterize.

Anaerobic reduction assays showed that sulfite reduction goes through enzymes that use NADH or NADPH as cofactors. Meanwhile, selenite reduction seems to prefer NADPH and tellurite reduction to NADH, suggesting that a common reduction mechanism is somehow unlikely. Experimentally, there were some drawbacks to the *in vitro* assays; for instance, sulfite can only be measured indirectly through oxidation of NADH or NADPH, while in the case of selenite and tellurite, the generation of their elemental forms can be determined spectrophotometrically. However, neither an increase of NAD(P)H oxidation nor the absence of $Se^0$ or $Te^0$ strongly suggests chalcogenide formation. In addition to chalcogenide detection mechanisms, it is interesting that the highest sulfite reduction was achieved using NADH since sulfite reductase uses NADPH as a cofactor.

Previous studies with MF05 showed that this strain does not exhibit remarkable reducing activities for selenite or tellurite aerobically, since these were not greater than 3 U/mg and 5 U/mg, respectively [25]. However, when performing the anaerobic tests with MF05, we could visualize elemental forms of both selenium and tellurium that are characteristic red-orange and black precipitates, respectively. Little information is available related to reducing activity in anaerobiosis, and it would be interesting to get proteomic data related to NS synthesis and metal exposure in anaerobic conditions. Furthermore, it would be interesting to identify which proteins are involved in the process, since heated cell extracts (with denatured proteins) showed lower activity in most cases.

From the results obtained by the genome analyses against several strains and more detailed against *E. coli* BW255113, it was possible to identify that the greatest difference between both strains lies in those genes that are not assigned a function by these databases, so it remains to identify which genomic elements could be granting the MMR phenotype and the ability to reduce chalcogenites and metals. Regarding cytotoxicity related to MF05, considering the possible uses in biomedicine, genome analyses showed that *stx1*, *stx2* or *eaeA* genes were not present in this strain (unpublished results). However, two different hemolysins (*hlyA*) were identified. Previous studies have concluded that *hly*-positive but *stx*-negative environmental isolates; also exhibit a certain degree of cytotoxicity [32], which should be considered if using a whole-cell system for NP synthesis. Moreover, the identification of new genetic determinants

that are associated with the synthesis of inorganic NS could be an interesting platform for new nanofabrication processes of structures of technological interest.

There is little information about the mechanisms of NS formation in biological systems; however, it is widely known that several reductases can reduce gold, copper, silver, and tellurium to their elemental forms [6–9, 33–36], which we were able to see as single NS for Se, Te, Li, Ag, Au, and Cu in MF05 under anaerobic conditions. Alternatively, a recent publication shows that the co-expression of metallothionein and phytochelatin synthase, in an aerobically grown recombinant *E. coli* strain, are involved in the biosynthesis of several single-element or multi-element NP, both in whole-cell and *in vitro* assays [37]. With the increased interest in NP production by biological process these structures had and will allow new industrial applications, such as antimicrobial (Ag-Au) and anticancer activities ($Cu_2O$), catalysis (dye degradation (Ag-Au-Pd), reduction of 4-nitrophenol (Au-Pd)), electrocatalysis (Au-Pd-Pt), chemiluminescence detection of glucose (Au-Ag) and adenine (Pt/X; X: Cu, Au, Ag), among others [38–42]. Thus, further studies must be conducted related to the compatibility of the biosynthesized nanostructures in our study for possible biomedical or industrial uses; however, several publications have proposed or demonstrated that CdS, Te, S, Au, Ag, and Cu NPs show application in diverse fields such as electronics and photonics (ex. solar cells and sensors), (bio)remediation of soil and water, agriculture, bioimaging, antimicrobial therapy and/or anticancer treatment [43–56].

## Supporting information

**S1 Fig. Clusters of Orthologous Groups (COGs).** COG functions comparison for *E. coli* BW25113 (blue) and MF05 (orange).
(TIF)

**S2 Fig. Growth curves of *E. coli* BW25113 and MF05 in M9 or M9\* media.** (A) Aerobic, (B) anaerobic *E. coli* BW25113, and (C) Aerobic, (D) MF05 cells grown in M9 (black circle) or M9\* (inverted green triangle) medium. Each point represents the average of three independent trials ± SD.
(TIF)

**S3 Fig. SEM EDX of MF05 whole-cell samples after under anaerobic conditions treatment.** Whole cells were (A) left untreated, exposed to (B) only $CdCl_2$, (C) only $Na_2SeO_3$ or (D) only $Na_2TeO_3$.
(TIF)

**S4 Fig. Checkerboards fluorescence emission of $Na_2SeO_3$ and $CdCl_2$ or $K_2TeO_3$ and $CdCl_2$ under aerobic or anaerobic conditions for MF05 in M9\* medium.** Fluorescence was monitored with 230 and 640 nm of excitation and emission wavelength, respectively. $Na_2SeO_3$ + $CdCl_2$ in (A) aerobiosis or (B) anaerobiosis, and $K_2TeO_3$ + $CdCl_2$ in (C) aerobiosis and (D) anaerobiosis. The results represent the average of three independent trials.
(TIF)

**S5 Fig. Excitation and emission fluorescence scan of sulfite and cadmium checkerboards for *in vitro* synthesis of CdS QDs with NADH or NADPH by crude extracts of MF05 under anaerobic conditions.** Excitation (A and C) and emission (B and D) scan spectra of crude extracts of MF05 treated with $Na_2SO_3$ (S) and/or $CdCl_2$ (Cd) in the presence of NADH (A and B) or NADPH (C and D).
(TIF)

**S6 Fig. TEM and SEM EDX of controls whole-cell assays in MF05.** Whole cells were (A) left untreated, or exposed to (B) only $CdCl_2$ or (C) only $Na_2SO_3$.TEM (upper images) and SEM EDX (lower images) showed no formation of NP under these conditions.
(TIF)

**S7 Fig. SEM EDX of MF05 cell extract samples treatments under anaerobic conditions.** Cells extracts were treated with (A) $Na_2SO_3 + CdCl_2$, (B)$Na_2SeO_3$, (C) $Na_2TeO_3$, (D) $HAuCl_4$, (E) $AgNO_3$ or (F) $CuSO_4$.
(TIF)

## Acknowledgments

Dr. Vásquez passed away before the submission of the final version of this manuscript. Dr. Arenas accepts responsibility for the integrity and validity of the data collected and analyzed.

We would also like to thank Fabián Araneda from CEDENNA for his assistance with the electron microscope operation and Dr. Natalia Valdés from Universidad de Santiago de Chile, Facultad de Química y Biología for her assistance with bioinformatics analyses.

This paper is dedicated to our beloved professor, Dr. Claudio Vásquez (R.I.P.)†. He firmly taught us science and life values.

## Author Contributions

**Conceptualization:** Mirtha Ríos-Silva, Myriam Pérez, Claudia Silva-Andrade, Juan Marcelo Sandoval, Claudio Vásquez, Felipe Arenas.

**Data curation:** Mirtha Ríos-Silva, Myriam Pérez, Roberto Luraschi, Claudia Silva-Andrade, Felipe Arenas.

**Formal analysis:** Mirtha Ríos-Silva, Jorge Valdés, Juan Marcelo Sandoval, Felipe Arenas.

**Funding acquisition:** Esteban Vargas, Jorge Valdés, Claudio Vásquez, Felipe Arenas.

**Investigation:** Mirtha Ríos-Silva, Myriam Pérez, Roberto Luraschi, Esteban Vargas, Claudio Vásquez, Felipe Arenas.

**Methodology:** Mirtha Ríos-Silva, Myriam Pérez, Roberto Luraschi, Esteban Vargas, Claudia Silva-Andrade, Jorge Valdés, Claudio Vásquez, Felipe Arenas.

**Project administration:** Claudio Vásquez, Felipe Arenas.

**Resources:** Jorge Valdés, Claudio Vásquez, Felipe Arenas.

**Software:** Claudia Silva-Andrade.

**Supervision:** Juan Marcelo Sandoval, Felipe Arenas.

**Validation:** Esteban Vargas, Juan Marcelo Sandoval, Claudio Vásquez, Felipe Arenas.

**Visualization:** Esteban Vargas.

**Writing – original draft:** Mirtha Ríos-Silva, Claudio Vásquez, Felipe Arenas.

**Writing – review & editing:** Mirtha Ríos-Silva, Juan Marcelo Sandoval, Claudio Vásquez, Felipe Arenas.

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
