## [Decision Letter · Decision Letter 0]

24 Jan 2022

PONE-D-21-18759Anaerobiosis favors biosynthesis of single and multi-element nanostructuresPLOS ONE

Dear Dr. Arenas,

Thank you for submitting your manuscript to PLOS ONE. After careful consideration, we feel that it has merit but does not fully meet PLOS ONE’s publication criteria as it currently stands. Therefore, we invite you to submit a revised version of the manuscript that addresses the points raised during the review process.

We look forward to receiving your revised manuscript.

Kind regards,

Erika Kothe

Academic Editor

PLOS ONE

Journal Requirements:

4. We note that you are reporting an analysis of a microarray, next-generation sequencing, or deep sequencing data set. PLOS requires that authors comply with field-specific standards for preparation, recording, and deposition of data in repositories appropriate to their field. Please upload these data to a stable, public repository (such as ArrayExpress, Gene Expression Omnibus (GEO), DNA Data Bank of Japan (DDBJ), NCBI GenBank, NCBI Sequence Read Archive, or EMBL Nucleotide Sequence Database (ENA)). In your revised cover letter, please provide the relevant accession numbers that may be used to access these data. For a full list of recommended repositories, see http://journals.plos.org/plosone/s/data-availability#loc-omics or http://journals.plos.org/plosone/s/data-availability#loc-sequencing.

Additional Editor Comments:

The reviewer identified some shortcomings. Please specifically see to the comment on animal vs. bacterial models.

Reviewers' comments:

Reviewer's Responses to Questions

**Comments to the Author**

1. Is the manuscript technically sound, and do the data support the conclusions?

Reviewer #1: Yes

2. Has the statistical analysis been performed appropriately and rigorously? 

Reviewer #1: N/A

3. Have the authors made all data underlying the findings in their manuscript fully available?

Reviewer #1: Yes

4. Is the manuscript presented in an intelligible fashion and written in standard English?

Reviewer #1: Yes

5. Review Comments to the Author

Reviewer #1: Dear Author,

The manuscript is well written and has good applications. But there are few queries with respect to the large scale production.

Kindly clarify the doubts.

Queries:

1.what about the cytotoxic effect of using Bacteria in your experimental study?

2. What is the stability of the particle size of bacteria used for the study ?

3. since the author has concluded the materials can be used for industrial applications, in this regard if the particle is going to be used in pharmacy or drug preparation, what is rating of its biocompatibility.

Suggestion:

The in vivo studies is always in animal model not in bacterial samples. Kindly correct in the manuscript and update the datas.

Regards

Dr. Premkumar J.

6. PLOS authors have the option to publish the peer review history of their article (what does this mean?). If published, this will include your full peer review and any attached files.

Reviewer #1: **Yes: **Dr. J. PREMKUMAR

Associate Professor

---

## [Author Response · Author response to Decision Letter 0]

23 Feb 2022

Dear Plos One Journal team:

Thank you for your feedback in our work. According to the notification received, we modified our manuscript, and all the changes made were highlighted in a document. We complemented the whole work with the queries and comments from the journal, editor and reviewer described below:

Query 1: Please ensure that your manuscript meets PLOS ONE's style requirements, including those for file naming.

Answer 1: We checked all the files and format to comply with the requested format, including file naming.

Query 2: We note that the grant information you provided in the ‘Funding Information’ and ‘Financial Disclosure’ sections do not match. 

Answer 2: We completed the funding information with the following paragraph:

This work received financial support from FONDECYT (Fondo Nacional de Ciencia y Tecnología) Regular 1160051 (CV), National doctoral scholarship CONICYT (Comisión Nacional de Investigación Científica y Tecnológica) 21170508 (MR), support from USA1799 Vridei (Vicerrectoría de Investigación, Desarrollo e Innovación) 021943CV_GO Universidad de Santiago de Chile (MR, CV), Basal FB0807 CEDENNA (EV) and Centro de Genómica y Bioinformática, Universidad Mayor (JV) is also acknowledged.

Query 3: We note that you have stated that you will provide repository information for your data at acceptance. Should your manuscript be accepted for publication, we will hold it until you provide the relevant accession numbers or DOIs necessary to access your data. If you wish to make changes to your Data Availability statement, please describe these changes in your cover letter and we will update your Data Availability statement to reflect the information you provide.

Answer 3: We deeply apologize for this misunderstanding since I thought you were referred to the sequence data. We added to the cover letter that we don’t have the data in a repository information. We wish to change the Data Availability Statement.

Query 4: We note that you are reporting an analysis of a microarray, next-generation sequencing, or deep sequencing data set. PLOS requires that authors comply with field-specific standards for preparation, recording, and deposition of data in repositories appropriate to their field. Please upload these data to a stable, public repository (such as ArrayExpress, Gene Expression Omnibus (GEO), DNA Data Bank of Japan (DDBJ), NCBI GenBank, NCBI Sequence Read Archive, or EMBL Nucleotide Sequence Database (ENA)). In your revised cover letter, please provide the relevant accession numbers that may be used to access these data. For a full list of recommended repositories, see http://journals.plos.org/plosone/s/data-availability#loc-omics or http://journals.plos.org/plosone/s/data-availability#loc-sequencing.

Answer 4: We clarified the preparation method in Materials and Methods section, lines 140-144 and we uploaded the sequence to NCBI GenBank, with Reference Sequence: NZ_VCSI00000000.1. You can go directly through the following link, in which all contigs are available with their respective sequences:

https://www.ncbi.nlm.nih.gov/nuccore/1679187305

And also in here: https://www.ncbi.nlm.nih.gov/Traces/wgs/VCSI01?display=contigs&page=1

Query 5: Please review your reference list to ensure that it is complete and correct. If you have cited papers that have been retracted, please include the rationale for doing so in the manuscript text or remove these references and replace them with relevant current references. Any changes to the reference list should be mentioned in the rebuttal letter that accompanies your revised manuscript. If you need to cite a retracted article, indicate the article’s retracted status in the References list and also include a citation and full reference for the retraction notice.

Answer 5: All References were checked and corrected according to the format requested and the number 33 was recently added to the discussion section, suggested by the reviewer.

Additional editor comments

Query 1: The reviewer identified some shortcomings. Please specifically see to the comment on animal vs. bacterial models.

Answer 1: All the comments were kindly incorporated to the manuscript. We changed all the “in-vivo” terms for “whole-cell”.

Reviewers' comments

Query 1: what about the cytotoxic effect of using Bacteria in your experimental study?

Answer 1: Previous works found virulence genes in several E. coli environmental isolates. Specifically, stx1(Shiga toxin 1), stx2 (Shiga toxin 2), eaeA (Attaching and effacing protein) and hlyA (hemolysin). Then, we examined our genome data and found two hemolysin genes in MF05 strain. This point was discussed in the manuscript in lines 380-390, and the reference was added.

Query 2: What is the stability of the particle size of bacteria used for the study?

Answer 2: This is a very important point, and during our experimental procedures, we were able to see the stability of the particles size, since fluorescence properties of QDs were present for at least 10 weeks at 4° C. Fluorescence in QDs is directly related to their size, so it might be possible that the bioprocess itself confers biostabilizers that could be very interesting to characterize in future experiments. This point was added in the discussion in lines 352-356

Query 3: Since the author has concluded the materials can be used for industrial applications, in this regard if the particle is going to be used in pharmacy or drug preparation, what is rating of its biocompatibility.

Answer 3: This comment was discussed inside our group and is absolutely necessary to characterize biocompatibility to define in a specific way, which uses can have these nanostructures. A comment related to this point was added in the discussion section in lines 379-381.

Query 4: The in vivo studies is always in animal model not in bacterial samples. Kindly correct in the manuscript and update the datas.

Answer 4: We changed all the “in-vivo” synthesis terms for “whole-cell” synthesis, claryfing that these are bacterial samples in which cell cultures are used to synthesize nanostructures.

We appreciated the feedback, and we hope that our answers are satisfactory for all of you. 

Sincerely,

Dr. Felipe Arenas Salinas

Associated Profesor

Universidad de Santiago de Chile

---

## [Decision Letter · Decision Letter 1]

27 Apr 2022

PONE-D-21-18759R1Anaerobiosis favors biosynthesis of single and multi-element nanostructuresPLOS ONE

Dear Dr. Arenas,

Thank you for submitting your manuscript to PLOS ONE. After careful consideration, we feel that it has merit but does not fully meet PLOS ONE’s publication criteria as it currently stands. Therefore, we invite you to submit a revised version of the manuscript that addresses the points raised during the review process.

We look forward to receiving your revised manuscript.

Kind regards,

Erika Kothe

Academic Editor

PLOS ONE

Additional Editor Comments:

The reviewer has (re)identified serious topics that need to be considered. Please very carefully address each point, since otherwise, the paper cannot be considered for publication.

Reviewers' comments:

Reviewer's Responses to Questions

**Comments to the Author**

1. If the authors have adequately addressed your comments raised in a previous round of review and you feel that this manuscript is now acceptable for publication, you may indicate that here to bypass the “Comments to the Author” section, enter your conflict of interest statement in the “Confidential to Editor” section, and submit your "Accept" recommendation.

Reviewer #2: (No Response)

2. Is the manuscript technically sound, and do the data support the conclusions?

Reviewer #2: No

3. Has the statistical analysis been performed appropriately and rigorously? 

Reviewer #2: Yes

4. Have the authors made all data underlying the findings in their manuscript fully available?

Reviewer #2: Yes

5. Is the manuscript presented in an intelligible fashion and written in standard English?

Reviewer #2: No

6. Review Comments to the Author

Reviewer #2: The current script (PONE-D-21-18759R1) focused on an environmentally synthesis of single and binary nanostructures under anaerobiosis. The idea seems to be good, however, there are some problems have to treat for reconsideration. These comments have to address in the revised version:

1. Abstract is so weak and have to support with some data.

2. Authors neglect big survey for recent works for inorganic mono and bimetallic nanoparticles and their applications. These recent publications are quite important to cite [Biocatalysis and Agricultural Biotechnology 2022, 39, 102261; Surfaces and Interfaces 2021, 25, 101175; Carbohydrate Polymers 2021, 266, 118163; Journal of Molecular Liquids 2021, 321, 114669; Polymer Testing 2020, 89, 106720; Rsc Advances 2016, 6 (78), 73974-73985; International journal of biological macromolecules 2019, 138, 450-461; International Journal of Biological Macromolecules 2020, 156, 829-840].

3. Number of references in the last five years is less than 30% and this is not sufficient. Number of recent works has to increase for double at least.

4. Number of literatures has to increase. 33 references are quite low numbers.

5. Improve the quality of figures, as they are not clear enough. High resolution figures are required.

6. All TEM micrographs have to include the same bare size. Check Figures 4, 6, 7.

7. Size distribution of the synthesized nanoparticles has to measure.

8. FTIR is required for the synthesized nanoparticles.

9. Compare the size of nanoparticles with literature [Surfaces and Interfaces 2021, 25, 101175; Carbohydrate Polymers 2021, 266, 118163; Journal of Molecular Liquids 2021, 321, 114669; Polymer Testing 2020, 89, 106720].

10. Compare the fluorescence of quantum dots with CQDs in literature [RSC Advances 2020, 10 (70), 42916-42929; International Journal of Biological Macromolecules 2021, 170, 688-700; Journal of Colloid and Interface Science 2021, 604, 15-29; Cellulose 2021, 28 (15), 9991-10011].

11. Conclusion is so short and have to support with more information.

12. Check and revise the language in whole manuscript.

Major revision is required.

7. PLOS authors have the option to publish the peer review history of their article (what does this mean?). If published, this will include your full peer review and any attached files.

Reviewer #2: No

---

## [Author Response · Author response to Decision Letter 1]

30 Jul 2022

Dear Plos One Journal team:

Thank you for your feedback in our work. According to the notification received, we modified our manuscript, and all the changes made were highlighted in a document. We complemented the whole work with the queries and comments from the journal, editor and reviewer described below:

Requirements:

1. Abstract is so weak and have to support with some data.

-The abstract has been rewritten and updated as requested.

2. Authors neglect big survey for recent works for inorganic mono and bimetallic nanoparticles and their applications. These recent publications are quite important to cite [Biocatalysis and Agricultural Biotechnology 2022, 39, 102261; Surfaces and Interfaces 2021, 25, 101175; Carbohydrate Polymers 2021, 266, 118163; Journal of Molecular Liquids 2021, 321, 114669; Polymer Testing 2020, 89, 106720; Rsc Advances 2016, 6 (78), 73974-73985; International journal of biological macromolecules 2019, 138, 450-461; International Journal of Biological Macromolecules 2020, 156, 829-840].

-We update the reference section. We apologize since, this paper was submitted last year, peer-reviewed in February and now re-peer-reviewed. The publications recommended have been include as references.

3. Number of references in the last five years is less than 30% and this is not sufficient. Number of recent works has to increase for double at least.

 -We updated the reference list, with 20 works in the last five years. 

4. Number of literatures has to increase. 33 references are quite low numbers.

-We updated and increase the reference list, now there are 56 references. 

5. Improve the quality of figures, as they are not clear enough. High resolution figures are required.

 -We improved the resolution of the figures.

6. All TEM micrographs have to include the same bare size. Check Figures 4, 6, 

- It results us difficult to have the same bar size, since different precursors for nanostructure synthesis have been used, so the biological components related to synthesis might be different so as the sizes obtained. In addition, TEM micrographs were an external service so given the complexity related to all processes needed to recharacterize synthesis products, we could say we would not be able to obtain new TEM images. 

7. Size distribution of the synthesized nanoparticles has to measure.

-We made measurements by DLS, however the polydispersity index was very high, given that the biological method has many impurities. In addition, the nanostructures obtained are of an irregular type.

8. FTIR is required for the synthesized nanoparticles.

- We didn’t consider IR characterizations because these nanoparticles, obtained through biological processes using bacteria, were not purified so there might be a lot of organic components attached that would interfere roughly in this kind of characterization.

9. Compare the size of nanoparticles with literature [Surfaces and Interfaces 2021, 25, 101175; Carbohydrate Polymers 2021, 266, 118163; Journal of Molecular Liquids 2021, 321, 114669; Polymer Testing 2020, 89, 106720].

-Unfortunately, and given that the biological synthesis of nanoparticles is not a process directly comparable to chemical synthesis, mainly due to the presence of biological material (proteins or metabolites) which interfere with the calculation of size and direct comparison with nanoparticles or marked manuscripts.

10. Compare the fluorescence of quantum dots with CQDs in literature [RSC Advances 2020, 10 (70), 42916-42929; International Journal of Biological Macromolecules 2021, 170, 688-700; Journal of Colloid and Interface Science 2021, 604, 15-29; Cellulose 2021, 28 (15), 9991-10011].

-The spirit of this paper is not to compare fluorescence properties between different QDs, but to study reduction capacity of bacteria under anaerobiosis to produce nanostructures. Fluorescence properties of CdS were used as an indicator of nanostructures synthesis. However, in the future quantum yields and other studies might be conducted in order to compare QDs from different natures (carbon and chalcogenides).

11. Conclusion is so short and have to support with more information.

-The conclusion has been deleted, according to the Journal guidelines it is optional to be written.

12. Check and revise the language in whole manuscript.

-The whole manuscript has been checked and revised, in both grammar and orthography.

---

## [Decision Letter · Decision Letter 2]

9 Aug 2022

Anaerobiosis favors biosynthesis of single and multi-element nanostructures

PONE-D-21-18759R2

Dear Dr. Arenas,

We’re pleased to inform you that your manuscript has been judged scientifically suitable for publication and will be formally accepted for publication once it meets all outstanding technical requirements.

Kind regards,

Erika Kothe

Academic Editor

PLOS ONE

Additional Editor Comments (optional):

Reviewers' comments:

Reviewer's Responses to Questions

**Comments to the Author**

1. If the authors have adequately addressed your comments raised in a previous round of review and you feel that this manuscript is now acceptable for publication, you may indicate that here to bypass the “Comments to the Author” section, enter your conflict of interest statement in the “Confidential to Editor” section, and submit your "Accept" recommendation.

Reviewer #2: All comments have been addressed

2. Is the manuscript technically sound, and do the data support the conclusions?

Reviewer #2: Yes

3. Has the statistical analysis been performed appropriately and rigorously? 

Reviewer #2: Yes

4. Have the authors made all data underlying the findings in their manuscript fully available?

Reviewer #2: Yes

5. Is the manuscript presented in an intelligible fashion and written in standard English?

Reviewer #2: Yes

6. Review Comments to the Author

Reviewer #2: Authors considered my comments carefully and the manuscript is significantly improved. The revised version can be ACCEPTED in its form

7. PLOS authors have the option to publish the peer review history of their article (what does this mean?). If published, this will include your full peer review and any attached files.

Reviewer #2: **Yes: **Prof. Hossam Emam

---

## [Editor Report · Acceptance letter]

26 Sep 2022

PONE-D-21-18759R2 

Anaerobiosis favors biosynthesis of single and multi-element nanostructures 

Dear Dr. Arenas:

I'm pleased to inform you that your manuscript has been deemed suitable for publication in PLOS ONE. Congratulations! Your manuscript is now with our production department. 

Kind regards, 

on behalf of

Prof. Dr. Erika Kothe 

Academic Editor

PLOS ONE